# Nutrition and Digestive Physiology of the Broiler Chick: State of the Art and Outlook

**DOI:** 10.3390/ani11102795

**Published:** 2021-09-25

**Authors:** Velmurugu Ravindran, M. Reza Abdollahi

**Affiliations:** Monogastric Research Centre, School of Agriculture and Environment, Massey University, Palmerston North 4442, New Zealand; M.Abdollahi@Massey.ac.nz

**Keywords:** newly hatched chick, gastrointestinal tract, nutrition, broiler

## Abstract

**Simple Summary:**

The first week after hatch is the most challenging period in the life of broilers. The digestive tract of the newly hatched chick is immature and must undergo dramatic changes before it can efficiently digest and absorb nutrients. The gut is the vital organ where nutrient digestion and absorption take place. Ontogenic changes that accompany improved digestion and absorption include increased secretion of digestive enzymes, increase in the gut absorptive surface area, and enhanced nutrient transporters. The obvious limiting factors are the secretion and activities of digestive enzymes, and the surface area for absorption. These limitations are overcome as the birds grow older, with concurrent improvements in nutrient utilization. In addition, substantial changes also take place in the physical and functional development of the immune system and intestinal microbial ecology. However, the focus of the current review was on nutrition-related challenges and nutritional approaches to assist the chick during this highly demanding period.

**Abstract:**

Because the intestine is the primary nutrient supply organ, early development of digestive function in newly hatched chick will enable it to better utilize nutrients, grow efficiently, and achieve the genetic potential of contemporary broilers. Published data on the growth and digestive function of the gastrointestinal tract in neonatal poultry were reviewed. Several potential strategies to improve digestive tract growth and function in newly hatched chick are available and the options include breeder nutrition, in ovo feeding, early access to feed and water, special pre-starter diets, judicious use of feed additives, and early programming.

## 1. Introduction

Often a seemingly straightforward aspect of nutrition reveals itself to be, upon closer inspection, not so simple after all. An illustration of such a conundrum is the nutrition of the newly hatched broiler chick. Conceptualizing the different challenges faced by the hatchling and solving them should be remarkably simple, but the difficulty lies in the complexity in the development of the gastrointestinal tract (GIT), digestive physiology, immune system, and intestinal microbiome.

The period immediately after hatch is the most critical period in the life of a broiler chicken. When the chick emerges from the egg, its digestive and immune systems are still immature, and the bird is not prepared to face the challenges confronting it. The changes are not gradual but abrupt: first, the switch to aerial breathing; second, initiation of thermal regulation; and third, the transition from yolk lipid nutrition to oral nutrition of complex dietary constituents. Associated with these changes are the substantial physical and functional development of the GIT (and digestive organs) and the maturation of active immunity. In consequence, the capacity to digest the feed and absorb and transport nutrients is limited during the early life of broilers. To achieve the genetic potential of modern broilers, the neonate must quickly adapt to efficiently digesting and utilizing nutrients from complex exogenous dietary sources in which the energy is supplied predominantly by carbohydrates.

The poultry industry has advanced remarkably over the past 60 years. Among meat industries, poultry meat production has undoubtedly been the most successful. Production standards of broilers have continually improved over this period, with male broilers currently capable of reaching a live weight of 2.6 kg at 33–35 d of age. Genetic selection brought about by commercial breeding companies is responsible for the greater part of the improvements in broiler growth [1,2]. The need to achieve and sustain the improvements in genetic potential is the driving force behind the recent advances in poultry nutrition and there had been concurrent refinement in the nutrition and feeding of commercial poultry starting from the hatch. As the growing period of modern broilers continues to shorten, the early nutritional management of the chick becomes increasingly important to success. Today, the first week represents 20–25% of the total production period. Because of the relatively short life of a broiler chicken, most of the bird’s physiological systems are not mature even at the time of marketing. Therefore, managing the immaturity of the GIT in the first week of life is crucial to the overall productivity. The relative daily growth rate of broilers is high during the early growth phase. In older broiler strains, body weight was reported to increase by 14% per day on the first day after hatch, reaching a peak of 22% per day by day 11 [3]. Similar findings have been reported by Nir et al. [4], who calculated that broiler chickens achieved a maximal relative growth rate of 20% at 5 days of age, which was maintained until day 10 and decreased thereafter to 16% by 14 days of age. However, during the past 3 decades, the growth of broilers has increased. At hatch, the broiler chick weighs around 42 g, which increases to 175 g at day 7. This increase represents 19 g/day or 300% over the first week. Changes that accompany this post-hatch growth include phenomenal growth of GIT, increased secretion of digestive enzymes, increases in overall gut surface area for absorption, improved nutrient transport systems, and development of the immune system. A better understanding of these changes may allow a more prudent exploitation of the immature GIT.

Despite the large volume of investigations conducted on aspects of early nutrition of the newly hatched chick, there remains a number of unanswered questions. In some cases, the results are inconsistent, suggesting the need for further research to understand the management tools to maximize the resilience during the first week of life. Nutrient digestion and absorption are highly complex processes with various integral components exhibiting different developmental patterns of activity. Secretion and activities of different digestive enzymes as well as nutrient transport systems vary during development after hatch. As highlighted in a number of reviews [5,6,7,8,9], the newly hatched chick has not been fully assessed for some of these aspects and, based on conflicting published data, it is not possible to present an integrated developmental view of digestion after hatch.

An overview of the current understanding of the GIT growth, digestive capability, and digestion during the first 7 days of life is presented herein. Potential early nutrition strategies that may support maximum growth and efficiency during latter growth stages are examined and those research areas warranting further work are identified. There have been exhaustive reviews relating to the development of the digestive tract and early chick nutrition [9,10,11,12], which provide background information to the current overview.

The immune system of poultry is only partially developed at hatch [13]. The development of this system, particularly of gut-associated immunity, responds to early feeding and dietary nutrients and is critical for the protection against exogenous organisms during week 1. Excellent reviews of the ontogeny of the immune system in neonatal poultry are available [14,15] and will not be covered herein.

The role of the microbiome, or the lack of it, in the overall development of the newly hatched chick cannot be discounted. The complex microbial community of the gastrointestinal tract plays an important role in production by aiding the development of intestinal structure, digestion, protection against pathogens, and the maturation of the host immune system. The digestive tract of the hatchling is sterile [16] but rapidly colonized by microbiome via the feed and environment. A stable microbiome, with a high-species diversity and an even distribution of predominant species [17], appears to be established by the third week of life [18,19]. The intestinal microbiome contains various bacterial species that are heavy consumers of amino acids (AA) and energy for their growth and colonization. Thus, the absence or low bacterial population during week 1 may provide an advantage to the host in terms of nutrient utilization. A discussion of the developmental aspects of the microbiome in the hatchling is beyond the scope of the current review and the readers are directed to reviews by Apajalahti et al. [20], Oviedo-Rondon [21], and Yadav and Jha [22].

## 2. Role of Residual Yolk Sac

The lipids of the yolk represent the primary nutrient source for the chick embryo, providing over 90% of the energy required for development as well as supplying a range of structural components for membrane biogenesis [23]. Glycogen stored in the liver and muscles of the developing embryo is the main energy source during the pipping process at hatching [5]. During this step, glycogen is utilized to meet the high energy demand of the hatching and, consequently, glycogen reserves are markedly depleted at the end of incubation. Despite extensive utilization of yolk lipids during the last week of incubation, up to one-quarter of the lipids that are originally present in the egg yolk remain unutilized at the time of hatch [23]. This residual yolk sac is internalized into the body cavity on the 19th day of incubation through the umbilical opening [24]. After the yolk sac is withdrawn, the yolk stalk remains as an appendage of the distal duodenum. The presence of the residual yolk sac during the first 3 days after hatching is critical for the growth and development of chicks. Chamblee et al. [25] reported that a significant increase in body weight was recorded only after 20% of the residual yolk sac is absorbed.

In the newly hatched chick, there is a rapid uptake of residual yolk material and only a vestigial amount remains after day 4 [26]. Nir et al. [4] found that the yolk residue decreased rapidly from 11% of body weight at hatch to 2% by day 2 and was negligible by day 4. Nitsan et al. [27] showed that 75% of the residual yolk present at hatch is utilized by day 3 and that by day 6 vitelline residue had decreased to negligible levels. Iji et al. [28] reported the yolk sac to be 8% of chick weight at hatch, decreasing to less than 1% of body weight by day 7. Murakami et al. [29] found that deutectomizing the chicks had no effect on the metabolizability of dietary energy or lipids but delayed the growth by 2 days, highlighting the role of residual yolk in complementing the nutrients supplied by the feed. In this study, the yolk residue disappeared rapidly over the first 3 days and was almost completely utilized by 7 days of age.

When there is no feed access, the newly hatched chick can potentially use residual yolk lipids as the primary source of energy. Noy and Sklan [30] estimated that the yolk represents about 20% of the body weight of the hatchling and contains about 50% lipids, providing immediate energy post-hatch. Breakdown of lipids, by the lipolytic enzymes in the yolk sac, can provide more than 90% of the total energy required for the hatching process [31,32,33]. Murakami et al. [29] estimated that the residual yolk contributed for approximately 30% towards the total dietary energy intake during the first 3 days. Two alternative suggestions have been proposed for the assimilation of remnant lipids [23]: (1) absorption via the yolk sac membrane through direct release into the circulation of the chick and/or (2) expulsion through the yolk stalk into the gastrointestinal tract. There is also some suggestion that the presence of the residual yolk sac may be imparting beneficial effects on the utilization of protein [34] and energy [35]. However, the exact mechanism or contribution of the yolk sac towards nutrient utilization remains unclear [36].

It needs to be emphasized that the utilisation of residual yolk lipids for energy supply is a wasteful process as specific nutrients in the residual yolk are much more valuable functionally by providing maternal antibodies for passive immunity and phospholipids, choline, and triglycerides for cell membrane development. Such wastage can be eschewed by ensuring access to feed to the chick soon after hatching.

## 3. Growth and Development of the Gastrointestinal Tract

Once pipped out, the chick needs to quickly adapt from obtaining its nutritional requirements from the yolk sac lipids to utilizing a diet based mostly on carbohydrates. To meet the needs for growth and maintenance of the rapidly growing chick, the digestive system is required to digest and absorb the exogenous nutrients at a rate adequate to meet its demands. In consequence, the chick places high precedence on intestinal growth to ensure that the nutrient supply functions are met. In addition to the physical architecture of the GIT, a strong barrier function and the immune system must be in optimal condition. For an excellent summary of development aspects of the intestinal system in birds, the readers are directed to Dibner and Richards [10].

The growth of the digestive tract occurs allometrically, with components of the GIT growing at different rates than the rest of the body. In the days following hatching, weights of proventriculus, gizzard, and small intestine increase more rapidly in relation to body weight than other organs and tissues [37]. This growth is maximal between 4 and 8 days of age and thereafter there is a relative decline. The mass of the small intestine increases almost six times within the first 7 days. Uni et al. [38] observed that the relative intestinal weight increased four times from hatch to 4 d of age and that the maximum proportional digestive organ weights were reached between 3 and 8 days of age. The intestinal weights decreased from 7 days to 21 days of age. Iji et al. [28] reported that the relative weights of the GIT and digestive organs exceed that of body weight gain during the early period of life and that the peak intestinal weight was achieved between days 7 to 14. The length of the small intestine and its individual component segments also increase with age [39].

Nitsan et al. [27] found that the relative weights of duodenum, jejunum, and ileum reached a maximum at 6 days of age and declined thereafter. Similar results were obtained by Nir et al. [4], with maximum relative weight of the small intestine occurring at 5 days of age, and by Nitsan et al. [3], who showed a maximum relative growth rate 4-fold that of body weight gain at 8 days of age. Iji et al. [28] and Murakami et al. [29] also showed the relatively small intestinal weight to be maximal at 7 days of age and declining thereafter. Ravindran et al. [40] found that the relative weights of intestinal segments (duodenum, jejunum, and ileum) were maximal during wks 1 and 2 of life and declined rapidly thereafter. These observations lend support to the premise that accelerated development of the supply organs immediately after hatch is a prerequisite for the sustained post-hatch muscle growth in fast-growing broilers. However, the intestinal mass, measured as g tissue/cm tissue, steadily increased from hatch to 35 days of age. This finding indicates that, though the relative size of the intestine declines with age, this decline is compensated by increased intestinal mass to support the nutrient supply function to the demand tissues.

The gizzard is known as the ‘pace-maker’ of normal gut motility. In addition to contributing to the grinding action, an increased activity of the gizzard allows greater gastric and/or intestinal refluxes [41], thus improving mixing of digesta with enzymes and nutrient digestion. At the time of hatch, the gizzard is the largest organ associated with the GIT and even larger than the liver (52 vs. 33 g/kg body weight). However, the relative weight of the gizzard steadily declines with advancing age [40].

Nitsan et al. [3] showed an increase in the relative pancreas weight to 8 days of age, at which stage it had an allometric growth rate approximately 4-fold that of body growth. After 8 days, the rate decreased and by day 23 the allometric growth rate of the pancreas was 1.5 times that of body weight. In this study, the liver reached a maximum allometric growth rate of two on day 11 and by day 15 this declined to be similar to that of body growth. In a subsequent study, Nitsan et al. [27] showed that the relative liver and pancreas weights peaked at 6 and 9 days of age, respectively.

Pinchasov [42] showed that the relative weights of the GIT, liver, and pancreas of broiler chicks increased in the first 24 h after hatching regardless of whether the birds had been fed, although the increase was greater in fed birds. A similar pattern of organ growth was observed in broiler chicks by Murakami et al. [29].

## 4. Maturation of Intestinal Mucosa

The functionality of the GIT is strictly related to its microscopic structure. The architecture of GIT is not well developed during the first week of life but rapidly matures with advancing age. The dramatic post-hatch increases observed in the weight and length of the small intestine could be considered trivial when compared to the growth of gut mucosa [43]. The changes in villus height, crypt depth, and submucosal thickness contribute greatly to the uptake of nutrients to meet the demands by the newly hatched chick. Increasing length and diameter of intestinal segments will also greatly influence the surface area available for absorption [40].

Uni et al. [44] found that the villus height and area increased rapidly at different rates in the three intestinal segments by 25–100% between days 4 and 10 posthatch and the increases were particularly evident in the jejunum and ileum. Crypt depth, which reflects enterocyte activity, increased until day 10. Maturation rate also increased linearly in these segments until day 10. It was observed that the height and perimeter of villi increased by 34% to 100% in all small intestinal segments between 4 and 10 days. The crypt depth and enterocyte number per villi also increased with advancing age. Similarly, Uni et al. [45] reported that the jejunal villus and crypt development occurs rapidly in the 4 to 5 days following hatch, with most epithelial cells proliferating at this point. Noy and Sklan [5] reported that the greatest duodenal villus growth rate occurred around or before 4 days of age, whereas the jejunal and ileal villi growth rates were maximized at day 10.

Uni et al. [38] observed that the villus volume in the duodenum, jejunum, and ileum did not change during the first 2 days after hatch, but rapidly increased thereafter. In the duodenum, this increase was complete by day 7, whereas the jejunal and ileal villus size continued to increase until day 14. The magnitude of increment with age was greatest in the duodenum and least in the ileum. Enterocytes’ numbers per villus were similar in all segments and changed little with age. Crypt depth increased 2- to 3-fold with age and was greatest in the duodenum.

Iji et al. [28] found that, although intestinal mucosa was structurally present at hatch, it matured rapidly with age through initial rapid cell proliferation, hypertrophy, and an increased rate of migration. The rate of cell proliferation peaked at 7 days of age and cellular migration peaked at 14 days of age. The increase in cell proliferation may be to support the growth of both the crypt and villus. Geyra et al. [46] found all cells along the villus in all segments of the intestine to be proliferating at hatch. Cell proliferation was sensitive to lack of feed, but, following re-feeding, cell proliferation was rapidly enhanced.

Continued genetic selection for faster growth has been accompanied by changes in the development and architecture of the GIT. Yamauchi and co-workers at Kagawa University, Japan [47,48], examined the development and maturation of intestinal segments of different lines of chickens in a series of studies. Their work confirmed the marked differences between fast-growing broilers and slow-growing layer chicks, with broiler chicks having intestines of greater length, weight, and surface area. Uni et al. [49] reported similar differences between heavy (Arbor Acres) and light (Lohman) strain chicks in intestinal development. Uni et al. [44] determined the posthatch intestinal morphology of these two strains in parallel with enzyme secretion, passage time, and digestion. Villus volume and enterocyte density were greater in the heavy than light strain at hatching and the rate of change with age was similar in both strains. Enzyme secretion per gram of feed intake into the duodenum was higher in the heavy strain on day 4 after hatch but no differences were apparent thereafter. Retention was 50% shorter in the light strain on day 4 but the difference was not significant from Day 10. Other researchers [27,50] reported similar differences and trends in enzyme secretion between low and high body weight lines.

A measure of cell proliferation within the mucosa can be obtained through the ratio of RNA to DNA, which gives an indication of ribosomal activity [49]. Uni et al. [51] found the RNA:DNA ratio to be higher in duodenal and jejunal tissues when compared to the ileal tissue, indicating greater cell proliferation. In all segments, the ratio decreased with age but at different rates. Protein-to-DNA ratios, which reflect cell size, were initially higher in the duodenum but decreased with age, indicating increasing cell size. The tissue activity, ribosomal activity, and cell size in all segments decreased with age but at different rates. Generally, duodenal tissue had the highest activity and ileal tissue had the lowest.

In summary, the growth of gut mucosa through cell proliferation and cell hypertrophy is rapid during the first week of life. The rate of increase varies between intestinal segments, but all reach the maximum rate of development between days 7 and 14.

## 5. Gastric pH

Digestive enzyme activity and microbial population are influenced by intestinal pH and any changes, therefore, will impact on digestive capability. Hydrochloric acid is necessary to maintain the low pH in digesta for the conversion of pepsinogen to pepsin, the enzyme initiating protein digestion [52]. Mahagna and Nir [53] found that the pH measured in the crop, gizzard, and small intestine declined from day of hatch to its lowest value on day 7, before increasing to a peak on day 14. No subsequent changes were observed in the pH of crop or small intestine. Barua et al. [54], in a study with wheat- and sorghum-based diets, reported a reduction in gizzard pH from d 7 to 14, followed by an increase from d 14 to d 42. Although the digestive secretions and concurrent secretion of the acid are expected to increase with advancing age, the pH increase beyond day 14 is probably a reflection of increasing consumption of feed with neutral pH, outstripping any influence of hydrochloric acid. The amount and size of limestone used in feed formulations is also a major determinant of intestinal pH [55].

## 6. Secretion of Bile and Digestive Enzymes

Digestion and absorption of nutrients in the GIT is a two-stage process, involving enzymatic breakdown and transport of products across the intestinal epithelium. Available data of the transport mechanisms for different nutrients during early life, which are not exhaustive [56,57,58], offer contradictory speculations regarding the sufficiency of transport systems during early life. Obst and Diamond [59] stated that transport systems do not limit early growth and that transport capacities are generally regulated to match or slightly exceed nutrient inputs. On the other hand, Croom et al. [60] were of the opinion that glucose transport capacity may be limiting, particularly in birds selected for rapid growth.

Hence, the discussion below focuses mainly on patterns in enzyme secretion with age. From day 17 to 21 of incubation, the intestinal secretion of sucrose, isomaltase, and aminopeptidase increases substantially [61]. Nevertheless, at hatch, the chick still has only a limited ability to digest proteins and lipids [62,63].

### 6.1. Biliary Secretions

The digestion of lipids is unique and differs from that of other major nutrients in that it needs be emulsified before it can be hydrolyzed by the enzyme, lipase, and absorbed [64]. The emulsification step requires adequate amounts of bile. Bile, produced by the liver, is composed of bile acids and salts, phospholipids, cholesterol, pigments, water, and electrolytes [65] Noy and Sklan [66] found that the secretion of bile components, including bile salts and fatty acids, into the duodenum increased 8- to 10-fold between days 4 and 21 post-hatch. However, the secretion of bile during week 1 is thought to be limited and responsible for the poor fat absorption [67].

### 6.2. Pancreatic Enzymes

Secretion of pancreatic enzymes, namely, trypsin, chymotrypsin, amylase, and lipase, is altered in response to feed consumption and dietary composition. For example, the neonatal chick is reported to respond to adjustments in starch intake by increasing the amount of amylase secreted [68]. Noy and Sklan [66] measured the secretion of lipase, trypsin, and amylase secretion in the duodenum from 4 to 21 days of age and also measured the total nitrogen secretion as an indicator of total enzyme secretion. Lipase, trypsin, and amylase secretions increased 20- to 100-fold during this period. Amylase secretion was relatively low at 4 days of age and increased rapidly with age. Nitrogen secretion increased by 15-fold between 4 and 21 days of age.

Noy and Sklan [66] measured the net secretion of amylase, trypsin, and lipase using 141Ce between days 4 and 21. Observed relative increases were lowest for lipase and highest for amylase, with increase in daily secretion of 100, 50, and 20 determined for amylase, trypsin, and lipase. These researchers also determined the changes in digestive capability over the first 21 days and concluded that the enzymes were secreted in adequate levels for fat and starch digestion but may not be sufficient for proteolysis in the early posthatch period.

Nitsan et al. [3] showed that the specific activity (units/g) of all pancreatic enzymes decreased during the first 3 to 6 days after hatching and then increased to 10% to 20% higher than at hatching on days 14, 11, and 21. Chymotrypsin activity increased from shortly after hatch to day 14. When expressed as units of activity per kilogram of body weight, the activity of all enzymes increased with age, reaching a maximum on day 8 for amylase and lipase and day 11 for trypsin and chymotrypsin. Nir et al. [4] observed that the specific activity of amylase was highest at hatch and decreased up until day 8. Lipase activity increased from a very low level at hatch by about 40-fold on day 14. Trypsin-specific activity increased gradually to reach a peak at 11 days of age, while chymotrypsin declined during the first 8 days of age and increased markedly thereafter. When expressed as units of activity per kilogram of body weight, the activity of all enzymes increased with age, reaching maxima for amylase on day 5 and for trypsin and chymotrypsin around day 11.

### 6.3. Brush Border Enzymes

The final step in digestion of dietary carbohydrates and proteins occurs on the surface of small intestinal enterocytes, in the immediate vicinity of the transporters that will carry the resulting sugars and AA into the epithelial cells. The enzymes responsible for this terminal stage of digestion (glucosidases, peptidases, and phosphatases) do not occur freely in the intestinal lumen, but rather in the plasma membrane of the enterocyte, and these embedded enzymes are referred to as brush border enzymes. Mahagna and Nir [53] measured the activity of two glucosidases, namely, saccharase and maltase, in broiler chicks and observed that the activities of both enzymes declined significantly from hatch to 7 days of age, whether expressed as units/g of tissue or units/kg body weight. Activity levels of both enzymes remained low until 21 days of age. A study by Uni et al. [38] reported that jejunal sucrase and maltase activities reached a maximum on days 1 and 2 after hatch and decreased thereafter. It was also reported that the mucosal sucrase and maltase activities were lower in the duodenum than in the jejunum or ileum. Iji et al. [69] reported similar results, with the maximal specific activities of sucrase and maltase at hatch and declining thereafter. However, they found that the total enzyme activity increased with age in all intestinal segments due to increased villus surface area and intestinal length. These authors found that the total enzyme activity per villus in the duodenum was higher than the activity per villus in the jejunum and ileum as a result of the longer villi in the duodenum.

Overall, perusal of available literature indicates that, although the trends in the secretion and activities of pancreatic and brush border enzymes with age vary between enzymes and studies, there is undeniable evidence suggesting that the pancreatic-specific enzyme activity (units/g) either decreases or remains stable over the first week of life. Although the total enzyme production (units/kg body weight) increases, the GIT grows at a relatively faster rate than body weight and feed intake increase over this period outpaces the secretion, confounding the elucidation of the enzyme data in hatchlings. The units used to express the secretion and activity of enzymes and varying effects in the different intestinal segments add further layers of difficulty in the interpretation of data.

## 7. Digesta Passage Rate and Viscosity

Digesta passage rate or retention time has a major influence on the digestion and absorption of nutrients. The slower the passage rate, the longer will be the digesta retention in the GIT, allowing more time for contact between digestive enzymes and substrates as well as products of digestion and intestinal mucosa. Noy and Sklan [66] found that the feed consumption increased 3-fold between 4 and 10 days of age and that this paralleled a 30% decline in passage time. After 10 days of age, however, no further change in passage time was observed, although feed intake continued to increase. The decrease in passage time was especially high in the duodenum, the major site of secretion of digestive enzymes, with a decline from 10 min on day 4 to 3 min on day 7, after which there was no significant change. Passage time through the small intestine declined from 161 min on day 4 to 110 min on day 14. Uni et al. [49] also observed a decrease in passage time during the first week. Passage time through the small intestine decreased from 115 min on day 4 to 74 min on day 7. However, in this study, passage time increased again by day 10 and reached 122 min on day 14. Barua et al. [54] measured the jejunal digesta viscosity at weekly intervals (day 7 to 42) in broilers fed wheat- and sorghum-based diets; the lowest digesta viscosity was recorded on day 7 and increased with advancing age, which could be a reflection of increased feed intake.

Over the first 2 weeks of life, the relative GIT size and feed intake markedly increase, but feed intake per unit of body weight declines [4,49]. It is suggested that the very rapid growth in gut size and feed intake may lead to short-term mechanical inefficiencies during the transit and mixing of digesta or both, influencing nutrient digestion [35].

The passage of digesta through the GIT is further complicated by episodes of reverse peristalsis between different intestinal segments [70]. Tur et al. [26] measured the GIT motility of young broilers (1, 8, and 15 days post-hatch) and observed that segments anterior to ileum increased their motility proportionally to broiler age, thus overcoming the slow passage issue to some extent in very young chicks. These motility patterns were attributed to the enhanced maturity of musculature and/or neuromotor system of the gizzard and increasing feed intake. It is established that the digesta retention time can be amplified by manipulating the feed particle size [71], but the usefulness of this approach to improve nutrient digestion during week 1 is unknown and seems to be dependent on grain type and hardness [72].

## 8. Digestion and Utilization of Nutrients

The outcome of the complex developmental changes in the anatomical structure, physiology, and functionality of GIT in the newly hatched broiler chick is reflected in the digestibility of nutrients and energy utilization. It is noteworthy that most of the published data on nutrient digestibility in very young chicks was measured over the total GIT, which could be confounded by urinary nutrient contribution and the modifying action of hind gut bacteria [73]. For these reasons, digestibility measurements at the ileal level are preferred but pose difficulties due to the low feed intake in young chicks leading to inadequate amounts of ileal digesta being collected for laboratory analyses.

A number of studies has investigated the digestibility of nutrients during the first few weeks posthatch. Starch digestion is thought to be not limiting in the newly hatched chick [74,75]. Chicks hatch with some reserves of amylase, which accumulates in the pancreas during the last few days of embryonic development [4,61], and are well adapted to starch digestion at hatch. Amylase activity seems to mature more quickly than other digestive enzymes. Despite the yolk containing less than 1% carbohydrates and the newly hatched chick never having ingested any feed, the intestinal mucosa at hatch contains a high level of disaccharidase activity [38]. Noy and Sklan [66] found that the net duodenal secretion of amylase, trypsin, and lipase was low at 4 days and increased 100-, 50-, and 20-folds, respectively, by 21 days. Total tract starch digestibility rapidly increased during the first days after hatch and a value of 97% is reached by days 4 and 8 of life in layer-type and broiler chicks, respectively [76]. High starch digestibility of 85–95% has also been observed in young chicks in other studies [66,77]. However, this putative view of near-perfect digestibility of starch at hatch could be challenged on the basis that it was measured over the total tract and could possibly have been affected by the modifying action of hind gut bacteria. On the one hand, as mentioned earlier, there is the difficulty of obtaining adequate amounts of ileal digesta during the first few days posthatch. Equally importantly, it must be noted that the steady-state passage conditions required for the measurement of ileal digestibility using indigestible markers are reached only after day 4 posthatch [9] due largely to the very low feed intake.

Noy and Sklan [66], using 141Ce as the nonabsorbable reference substance, measured the intestinal absorption of nitrogen, fatty acids, and starch from 4 to 21 days of age. Over this period, nitrogen absorption increased from 78% to 92%. However, starch and fatty acid absorptions were unaffected. The ileal starch and fat digestibility at 4 days of age was determined to be over 95% and 85%, respectively. These researchers opined that the digestion of starch and lipids is not a limiting factor to the growth of young chicks. In a recent study from our laboratory (unpublished data), investigating the age effect on starch digestibility of four common cereals (maize, wheat, sorghum, and barley) in broilers, the average digestibility of starch declined from 0.989 at d 7 to 0.958 at d 42. A highly significant negative correlation (*p* < 0.001; r = −0.782 to −0.921) was also observed between feed intake and starch digestibility. In contrast, Zelenka et al. [78] reported that the digestibility of lipids decreased from hatch up to 8 days of age and then gradually increased until day 14. Carew et al. [79] fed diets containing 20% fat sourced from maize oil or beef tallow and measured fat digestibility during days 2–7 and 8–15 post-hatch. The average digestibility of the maize oil increased from 85% to 95% and the beef tallow digestibility increased from 40% to 79% between the two assay periods. Of interest was the trend for the excreta fat to increase from day 3 to days 5 and 7, before decreasing to day 15. These responses were attributed by the authors to changes in the transit time. Tancharoenrat et al. [67] similarly found that the digestibility of fat is low in week 1 and that the digestibility of fats with a high proportion of saturated fatty acids was lower than those with high proportions of unsaturated fatty acids (Table 1). The ability to digest both saturated and unsaturated fats increased rapidly with age. It was speculated that the poor fat digestibility during week 1 is likely due to poor emulsification resulting from low bile secretion, inefficient recycling of bile salts, and/or inadequacy of fatty acid binding protein [80].

In general, peptic and pancreatic proteases exhibit increasing activities with age [81], leading to increased protein digestion after week 1. Batal and Parsons [82,83,84] measured the total tract amino acid digestibility of several diet types for chicks from hatch to 21 days of age in a series of trials. A trend for digestibility to increase with age was observed in all trials. Barua et al. [85] reported that the basal ileal endogenous flow of nitrogen and AA decreased quadratically by broiler age, with flows being higher on d 7, then decreasing on d 14, plateauing until d 35, and decreasing further on d 42. In a follow-up study with wheat- and sorghum-based diets [54], the age effect on AA digestibility was reported to be variable depending on the grain type and specific AA. The apparent digestibility of nitrogen and AA increased with advancing age in wheat but was unaffected by age in sorghum. When standardized for the age-appropriate basal endogenous AA losses [78], no age effect was noticed on the standardized digestibility of AA in wheat; however, the standardized AA digestibility values for sorghum were higher at day 7, reduced at day 14, and then plateaued. The different patterns of apparent vs. standardized digestibility values highlight the importance of considering age-specific endogenous AA flows in standardizing the AA digestibility, especially in young broiler chicks.

Published data on age effects on mineral utilization in broilers are limited. Thomas and Ravindran [86] measured the total tract retention of five major minerals (calcium [Ca], phosphorus [P], potassium, sodium, and magnesium) and four trace minerals (iron, manganese, zinc, and copper) on days 3, 5, 7, 9, and 14 of age. The retention of individual minerals differed widely and the retention of major minerals was found to be considerably higher than those of trace minerals. Age effects were significant for all minerals, except Ca. In general, retention was highest at day 3, declined to day 7, and remained unchanged until day 14 (Table 2).

Interestingly, similar declining trends with broiler age were observed for the ileal Ca digestibility of limestone [87]. Ileal Ca digestibility declined with advancing age, with the highest values being determined at days 7 and 14 compared to later stages. The ileal digestibility of Ca in limestone at days 7, 14, 21, and 42 were 51%, 53%, 36%, and 27%, respectively. It would appear that Ca absorption is more efficient during the first 2 weeks due to (1) the greater demand on the intestines to absorb more Ca and (2) possibly to compensate for the low feed and Ca intakes to meet the needs for rapid bone formation.

### Energy Utilization

Energy is not a nutrient, per se, but a property of energy-yielding nutrients (carbohydrates, lipids, and protein). The energy derived from carbohydrates, lipids, and protein is different, with lipids providing 2.5 times more energy than carbohydrates. These differences may explain part of the contradictory findings on the age effects on energy utilization. Two studies were conducted by Zelenka [34] to investigate the changes in the ability of hatchlings to metabolize energy. The first study found that the apparent metabolizable energy (AME) of a practical diet decreased rapidly from day 3 posthatch to reach a low point at 6–9 days before increasing again to 14 days of age. At 14 days of age, the dietary AME was 10% higher than that on day 9. A follow-up study showed a similar pattern of decrease from hatch to 7–8 days of age followed by a steady increase in AME to 14 days of age. In this study, the AME was 7% higher at day 14 than at day 8. Murakami et al. [29] also found that metabolizability of dietary energy and absorption of dietary lipids were highest at hatch, then declining to their lowest at days 5–6, and gradually increasing thereafter.

A series of trials were conducted at the University of Illinois to measure the changes in AME of maize–soybean meal diets over the first 14 days of age of broiler chicks. The first study [82] showed a linear increase in AME from day 2 to day 14, with nitrogen-corrected AME (AMEn) values of 12.43 MJ/kg on day 2 increasing to 14.35 MJ/kg on day 14. The second study [82] reported AMEn of 13.23 MJ/kg on days 0–2, which decreased to 12.55 MJ/kg on days 3–4 before increasing again to 13.52 MJ/kg on day 14. The third study [83] observed that AMEn values were higher at 0–2 days of age (12.67 MJ/kg) than at 3–4 days of age (11.41 MJ/kg) and then increased again by day 14 (13.49 MJ/kg).

Thomas et al. [35] observed that the AME of a maize–soybean meal diet was highest at day 2 posthatch, decreased at day 4, and then plateaued between days 6 and 10 before increasing at day 14 (Table 3). The surprisingly high AME estimates determined during the first few days posthatch and the decline during days 6–10 are interesting findings and consistent with previous reports [29,34]. The remarkably high nitrogen retention at day 2 was also noteworthy (Table 3) [35]. Khalil et al. [88] reported that broiler age had a substantial impact on the AMEn of maize, wheat, sorghum, and barley, and the effect varied depending on the cereal grain. In general, the highest AMEn values for all grains were recorded on day 7 and declined, either linearly or quadratically, with advancing age. These anomalies may be attributed, as discussed previously, to the presence of residual yolk sac, low feed intake and consequently inefficient feed passage during the first week post-hatch, inefficient feed passage, and the essentially sterile intestinal microbiome [35]. The intestinal microbiome is a heavy user of energy and nutrients [89] and the absence or low microbial population in the neonatal chick may, in part, provide the observed apparent advantages in terms of nutrient utilization and AME.

## 9. Development of Skeletal System

Bones are poorly mineralized at the time of hatch. Rapid mineralization and growth of bones occur during the first 2 weeks (when adequate Ca and P diets are fed), regardless of the growth potential of the chick. The residual yolk sac plays an important role in the supply of Ca during the first few days [90]. Calcium absorption is more efficient during the first 2 weeks, reflecting the greater demand on the intestines to absorb more Ca to meet the needs for rapid bone formation. At hatch, the fat-free tibia ash content is around 28% and increases to 42% by day 7. When there is normal mineralization, the fat-free tibia ash content is around 50%, which is reached only at 14 days (Table 4). These data highlight the very high demand for Ca during the first 2 weeks, coupled with more efficient Ca absorption to meet the needs for rapid bone formation. Similarly, Skinner and Waldroup [91] found that the percentage increase in tibia Ca concentration in broilers was greater during the first week compared to those at other ages (up to 8 weeks).

## 10. Physiological Limitations in the Newly Hatched Chick: Summary

The intestine represents only 3–4% of the total body mass of broilers during the first 2 weeks of life [40], but it is the most demanding organ in the body in terms of energy and protein needs. Any changes in intestinal growth and its metabolic demands are likely to influence bird performance. Available literature demonstrates that the relative growth rate of digestive organs in broilers is allometrically maximized within the first week of life and declines thereafter to eventually approach that of the gain in body weight. These findings are consistent with the demand on supply organs imposed by the maximal relative body growth being achieved from 5 to 10 days of age.

In the main, there is consensus that starch is well digested and that the digestion of fat and protein and metabolizability of energy are compromised in the newly hatched broiler chick. The AME and the digestibility of lipids and protein are low during the first 10 days of life and increase thereafter. But the changes over the first 14 days of life are not linear and the digestibility of some nutrients may decrease over the period of 5 to 9 days of age before increasing again by day 14. Contrary to the trends observed for major nutrients, mineral absorption and utilization are greatest during week 1 and decline thereafter, possibly reflecting their higher metabolic needs. The low digestibility of major nutrients in the hatchling is attributed to the poor development and maturation of the GIT. The digestive system is juvenile at hatch and its capacity to digest the feed and absorb nutrients is limiting. The digestion and nutrient absorption depend on a well-developed GIT with sufficient digestive enzyme secretion and developed intestinal morphology. Growth and development of the GIT of the broiler are rapid and exceed that of the body weight during the 2 weeks. In addition, the secretion and activities of lipase, amylase, and proteases, which are responsible for the enzymatic digestion of major nutrients, increase during the first 14 days of age. The villus height and crypt depth, which are the direct representation of the absorptive surface of the small intestine, also increase between 4 and 10 days of age.

## 11. Potential Strategies to Overcome the Physiological Limitations

Overall, the evidence suggests that intestinal growth and function of newly hatched chicks are not adequate to support efficient muscle development and growth. Thus, untapped opportunities exist for the realization of the modern broiler’s genetic potential through nutritional manipulation of digestive capacity during the critical first few days after the hatch. An enhancement of early intestinal growth even by a day or two may have a significant impact towards improving the efficiency of the bird over its grow-out period. Several strategies that may be considered, individually or in combination, to assist the hatchling to overcome the underlying physiological limitations are discussed below.

### 11.1. Strategies Prior to Hatching

#### 11.1.1. Breeder Hen Nutrition

The simplest practical strategy to enhance the intestinal growth in the embryo is through the nutrition of breeder hens and the enrichment of biologically important nutrients in their eggs. There is no physical barrier to the transfer of most nutrients from the breeder hen diet to eggs, as evidenced by available literature [92,93,94,95,96]. The success and popularity of n-3 fatty acid-enriched eggs have paved the way to enrich eggs with other functional nutrients like, inter alia, conjugated linoleic acid, vitamin E, vitamin D, selenium, folic acid, and carotenoids. Cherian [95,97] showed that early exposure to lipids and essential n-3 fatty acids via hatching eggs can influence cell membrane fatty acids, immune responses, and the production of inflammatory mediators in the hatchling. Early exposure to essential fatty acids has metabolic roles over and above the influence on providing energy during embryonic growth. There is evidence that this has beneficial effects on the early growth of the newly hatched chick. Cherian [95] speculated that this influence may extend through the entire production phase of broilers. However, data on the effects of altering the nutrient composition of hatching eggs on the performance of progeny are equivocal [93,94]. In general, any improvement in posthatch growth beyond week 1 is yet to be demonstrated.

#### 11.1.2. In Ovo Nutrition via Hatching Eggs

In ovo injection of vaccines into the amnion during the late embryonic stage of hatching eggs is commonplace in commercial hatcheries. In comparison with the traditional method of broiler vaccination, in ovo injection offers a less stressful, faster, and more uniform delivery of vaccines to developing embryos. This technology could easily be transferred to the administration of highly digestible nutrients to enhance embryonic development and has been extensively researched by Uni and Ferket [11]. A myriad of nutrients such as dextrin, maltose, sucrose, AA, glycerol, L-carnitine, creatine pyruvate, salt, minerals and vitamins, individually or in combination, has been evaluated to date. In ovo nutrition aims at injecting the egg with one or more nutrients during the last stage of embryonic development (around 3 days before hatching). The basis of this strategy is that, after internal pipping, the embryo ingests the remaining amnion [98] and the presence of this protein-rich liquid, along with nutrients inoculated into that fluid, stimulates the development of the GIT.

In ovo injection of nutrients has been shown to increase, inter alia, embryo villus surface area at 3 days post-hatch [99], brush border carbohydrase activity [100], dietary carbohydrate absorption [101], and brush border nutrient transporter activity [57]. Uni and Ferket [11] observed that the intestine of in ovo-fed chicks at hatch is at a similar stage of development as 2-day-old chicks. In general, there are also growth benefits seen during week 1 and early stages. However, this early benefit is not always carried over the whole grow-out period. Any advantage in early growth is mostly lost as the bird grows, which may be due partly to the well-established phenomenon of compensatory growth in broilers [102].

Although the technology of in ovo feeding was patented almost 20 years ago [11], its commercial uptake has not been widespread, due mainly to the limitations imposed by the need for specialized equipment, time, and capital investment. Injection of nutritional solutions in practice is generally complex. Other reasons include lack of tangible benefits in terms of final market weights and feed efficiency and the possible detrimental effects on hatchability from the inoculation of some nutrients.

### 11.2. Strategies after the Hatching

#### 11.2.1. Early Access to Feed

Getting a good start for the chick in hatchery operations is vital for maximizing the survivability, health, welfare, and productivity. At the hatchery, chicks pip out at different times over a 36–48-h time frame and are usually removed from the hatcher when over 95% have emerged from the shell. Once removed from the hatcher, the hatchlings undergo a number of treatments and are then transported before being placed on the broiler farm. Thus, in practice, chicks could be deprived of feed and water for up to 72 h. Such feed deprivation, however, has detrimental effects on chick development and growth [11]. Chicks being transported also suffer from the stress of handling in the hatchery and during the transport, which affect the development further.

As indicated at the start, the yolk residue can serve as a nutrient and energy reserve in the absence of feed for up to 72 h posthatch [103]. Its innate function, however, is to be a supply of phospholipids for the formation of cell membranes in the intestine [30] and maternal antibodies to the bird. When the chick ingests feed, uptake and utilization of yolk is increased [103] due to the physical action of the gut drawing the yolk material into the intestine via Meckel’s diverticulum to support the intended roles.

Optimal GIT development of the hatchling is ensured through providing feed access immediately after hatch. The benefits of immediate feeding have been attributed to several effects [104]: improved nutritional maturity, stimulation of yolk utilization, enhanced GIT development, and long-term metabolic responses. These benefits are well accepted and the role in stimulating the intestinal development, digestive functions, immune system, and muscle growth are now recognized [5,105,106,107,108]. The options include provision of feed and water in the hatcher [109], chick transport boxes, or both, rather than delaying until the birds reach the farm. These approaches represent an immediate potential area for improvement by the industry. It is noteworthy, however, Deines et al. [110] observed that access to feed and water in the hatcher improved body weights until 28 days and this advantage was lost at 42 days. Immediate access had no influence on the processing yield, feed efficiency, or mortality.

#### 11.2.2. On-Farm Hatching

On-farm hatching is an innovative concept, developed in the Netherlands, wherein the eggs that have been incubated for 18 days are hatched directly on the farm, thus preventing the burden of any stress. The hatched chicks have instant access to water and feed, allowing for better development of the GIT and organs. This strategy results in healthier and more robust chicks that are more resilient to environmental pressures [111]. De Jong et al. [112] reported that on-farm-hatched chicks were heavier than traditionally hatched chickens until 21 d of age, but the advantage was lost thereafter. A tendency for improved feed efficiency for on-farm-hatched birds was observed at 1.5 and 2.0 kg body weights. Importantly, the results showed that the on-farm hatching might be beneficial for broiler welfare, as it reduced wet litter, foot dermatitis, and total mortality.

This technology is now commercially available through different Dutch suppliers offering specific systems (Nestborn (Exergen), One2Born (one2Born B.V.), Patio system (Vencomatic), and X-treck (Vencomatic), varying in labor requirements, ease of use, and investment.

#### 11.2.3. Special Pre-Starter Diets

Feeding special, highly digestible pre-starter diets during week 1 is justified and the development of these diets should take into account the intestinal and nutritional limitations of the chick. Two options are available. First, solutions of highly digestible sugars and free AA, B-complex vitamins, and organic acids (pH 3.5–4.0) could be offered during the first 48 h, especially if the chicks are under obvious stress. Second, specialized starter diets, based on high-quality and highly digestible ingredients, could be used. Some basic principles that may be considered in such diets are listed in Table 5.

Caution must be exercised to minimize the use of ingredients with mycotoxins that exceed recommended limits; details of these toxins are provided in an elegant review by Bryder [115] and they are comprised of several key GIT functions, including decreasing the surface area for absorption, modulation of nutrient transporters, and loss of the barrier function. Some mycotoxins facilitate persistence of intestinal pathogens and the likelihood of intestinal inflammation. Mycotoxins, per se, may not be the cause of intestinal health problems, but may predispose the chicks to one [115].

For a good discussion of the composition of specialized pre-starter and starter diets, see Barekatain and Swick [116]. Special pre-starter supplements (e.g., Oasis; Novus International, Inc., St. Louis, MO) are also commercially available [83,117] and there is limited evidence that feeding of this hydrated, low-fat, highly digestible protein and carbohydrate nutritional supplement during the first 48 h has beneficial influence on the growth and meat yield of broilers. Influence of specialized diets on week 1 broiler performance is variable and a critical gap remains in the scientific literature.

Predictably, these special diets will be expensive but cost-effective and practical because of the low feed intake during week 1. Considering the potential long term on broiler growth, immunity, and gut health, the use of these special pre-starter diets should not be considered as a cost but as an investment [6].

The various options suggested in Table 5 need not be incorporated in a single formula, but could be prioritized and tested on ‘trial and error’ bases, taking into account the local conditions, before implementation in the field. However, based on available literature, it is concluded that the proposed options lead to only transitory, short-term benefits with any advantage being generally lost with age. According to Bhuiyan et al. [118], immediate post-hatch feeding has a greater positive impact on performance than feeding high-quality diets following delayed feeding.

#### 11.2.4. Feed Additives

In recent decades, feed additives have become vital components in practical diets to maintain health status, uniformity, and production efficiency in broilers. Of the plethora of additives available, three groups are particularly influential in the early nutrition of chicks (Table 6). The first one relates to exogenous enzymes that enhance digestion capability [119]. In the context of young chick nutrition, digestion may be enhanced by provision of enzymes that supplement limited enzyme capability (e.g., lipase, protease) and degradation of specific bonds in ingredients not hydrolyzed due to lack of endogenous digestive enzymes (e.g., phytase, carbohydrases). The enzymes that may be useful and widely used by the industry are the carbohydrases that cleave the viscous fiber components and phytases that target the phytate complexes in plant-based ingredients [120,121,122]. More recently, a technically successful monocomponent protease became available [123]. Effective emulsifiers that enhance lipid digestion are also now on the market [80]. While the use of these additives has become commonplace, the research data have generally covered the 21- or 35-day growth phases and none have specifically addressed the influence during the first 7 days.

The second relates to the ban or restriction on the use of antibiotics at sub-therapeutic growth-promoting levels. Antibiotics have been used in the poultry diets for over 70 years as a prophylactic measure against pathogens and sub-clinical diseases and, by doing so, to improve growth [124]. Due to public concerns, the global poultry industry is entering an era of antibiotic reduction or production without antibiotics. The withdrawal of this preventive measure has serious implications for the productivity and health of birds, particularly at early ages. Some alternatives that are being considered are listed in Table 6.

Third, geneticists have done their share in developing strains of broilers that are capable of producing protein gain at greater efficiencies than ever before. The challenge for nutritionists is to sustain these improvements in genetic potential by refining the amino acid nutrition of poultry. In this context, the commercial availability of synthetic AA has enabled the use of digestible AA, rather than total AA, as the basis of feed formulations [125] and to more precisely meet the ideal amino acid profiles [126]. To date, however, there has been no research on aspects of amino acid nutrition focusing on the newly hatched chick. Amino acid nutrition is intricate because AA are not only the building blocks of protein, but also involved in an array of functions that are unrelated to skeletal protein deposition and growth [127] that may be relevant to chicks during the first few days. Whether the amino acid requirements and ideal protein ratios immediately after hatch are different from those of after week 1 are currently unknown and future work in these areas will be instructive. There is information, albeit limited, some specific AA (threonine, glycine, serine, and glutamine in chick growth) [128,129] may be expedient in chick diets and the availability of synthetic forms make it possible to better comprehend the potential role of these functional AA.

#### 11.2.5. Early Programming

It is widely held that the imprinting or programming during early life has a notable impact on the long-term growth, metabolism, and health of an organism. This is a fundamental process of life and restricted to a narrow window of ‘critical period’ in an animal’s very early life [130]. In the past, the term ‘imprinting’ was used interchangeably with ‘genomic imprinting’, an epigenetic mechanism defined as gamete-of-origin dependent modification of the genotype. However, the current definition of imprinted changes does not involve the germ line, and they are not inherited by the next generation [131].

Although only limited information exists specifically relating to the newly hatched chick, evidence allows us to conclude that it can be programmed to enhance its tolerance to immunological, environmental, or oxidative stress [132]. Nutritional programming immediately after hatch can also influence the utilization or requirements of nutrients, including energy, protein, fatty acids, and minerals [97,133,134,135], while some bioactive dietary components may imprint intestinal microflora colonization [131,136]. For example, Yan et al. [133] reported that conditioning of broilers on a diet low in Ca and P for 90 h post-hatch improved intestinal Ca and P absorption at 32 days of age and increased the gene expression for the mineral transporter proteins. Rousseau et al. [135] similarly observed broilers were able to adapt to early dietary changes in P and Ca and improve digestive efficiency in later stages via increase in mRNA levels of several genes encoding Ca and P transporters. The extent of compensation in growth performance and bone mineralization depended on the Ca and P levels in the subsequent diet. These data demonstrate that epigenetic imprinting and nutritional adaptation to low dietary Ca and P might indeed be possible and likely for other nutrients as well.

The microbiome of the GIT has an immense influence on the host. The relationship begins at hatch and evolves into a stable and resilient ecosystem characterized by diverse, commensal microorganisms with advancing age. Once established by the third week of life, it will be difficult to change this ecosystem. In this context, early stimulation by gut flora enhancers is relevant to influence the entire growth cycle of broilers. A number of effective additives is available (Table 6) and such programming should be initiated with the first feed.

Although feeding special conditioning diets immediately after hatch presents great opportunities, it is logistically difficult to accomplish in practice using current production systems because of the time-lag between hatch-pull and farm placement. However, recent progress in on-farm hatching, discussed earlier, offers a practical strategy to provide conditioning diets at hatch and could be a game changer in the future.

More research is warranted before conditioning strategies could become routine in the broiler industry. First, the mechanisms by which insults during a critical window of development have long-term effects, many weeks later, on metabolism of an animal are only starting to emerge [130], with evidence from humans and other mammals. Second, the challenge is to identify the critical time(s) when the chick still shows some metabolic and physiological plasticity. The time and duration of this period depend on the animal species and the physiological systems (digestive, absorptive, immune, and microbial) and whether the currently assumed 0–72 h posthatch is the appropriate phase need to be confirmed in future studies. In the case of microbiome interventions, based on fine-scale temporal dynamics of chicken GIT microbiome, recent work by Jurburg at al. [137] suggests that 3–4 and 14–21 days posthatch may be the optimal times for microflora interventions.

Taken together, it appears that metabolic and nutritional traits might be imprinted during the first few days post-hatch by adaptive conditioning of gene expression. Trans-generational transmissions of these traits in birds, however, are still anecdotal and yet to be investigated.

## 12. Final Thoughts

The nutrition of newly hatched chicks to improve the overall efficiency of broiler production is a topical issue. With anticipated genetic progress, the first 7 days may account for more than 25% of the life of broilers in the future. A good case in point is the current performance levels in New Zealand, where male broilers are already reaching a processing weight of 1.85, 2.60, and 3.25 kg at 28, 35, and 42 days, respectively. There are, however, biological and metabolic limits to the rate of growth, efficiency, and meat yield, and such faster growth may have unintended negative consequences. The adverse effects include welfare issues resulting from skeletal abnormalities, poor immune responsiveness, increased pathogen susceptibility, metabolic disorders, and meat quality issues such as wooden breast and white striping. Thus, the focus of early nutrition strategy should not be only on body weights, but also on eluding the above concerns, which have become key industry issues in recent years.

The apparent paradox is that years of interest and intensive research have not brought us closer to a better understanding of the physiological systems of the newly hatched broiler chick and of effective strategies to effectively manipulate broiler performance. Among the possible reasons for this lack of progress, four are worth mentioning. First is the inability to translate early improvements in growth to long-term benefits at market age, perhaps due to the compensatory growth. Time is opportune now to shift the focus from final body weight and investigate the influence of early nutrition on other parameters such as production efficiency, nutrient sustainability, immunity, metabolic disorders, and bird welfare. Second is the widespread use of low levels of in-feed antibiotics in poultry feeds as a growth promoter, a practice that may have partly masked the deficiencies in the maturation of digestive, immune, and microbial systems of the hatchling. Despite popular use since the 1950s, the exact mode of action of in-feed antibiotics remains a matter of conjecture, but it is thought to involve alterations in the balance and population of intestinal microbiome [124]. It follows that a better appreciation of the intestinal microbial ecology is vital. Estimates indicate only a small proportion of the bacteria is identified by traditional culture-based techniques [20,138]. The inability to fully characterize the microbiome is the third issue that limits the intervention strategies. While acknowledging the highly complex nature of the GIT microbiome, strategies cannot be successful unless the basic knowledge is available. The intestinal microbiome plays critical roles in the development and physiology of GIT, nutrient absorption, modulation of the immune system, and resistance to pathogen invasion. With the advent of molecular techniques, rapid advances are envisaged in the exploration of the intestinal ecosystem [139]. Fourth, significant progress in genetic selection has been made over the past three decades to improve the growth and body weights of broiler chickens. Along with faster growth, feed efficiency and the proportion of breast meat have improved, exacerbating metabolic, immune, and welfare challenges. Continually shortening the broiler growth cycle puts further pressure on potential strategies and not only restricts the worth of research conducted with older strains but may make research conducted with them less relevant. It must be emphasized that science is a continuously evolving process and the approaches must be constantly guided by the totality of evidence using modeling and meta-analysis instead of results from limited, individual studies.

The preceding discussion provides some background understanding on the theme of nutrition of the newly hatched broiler chick; however, as highlighted, much remains to be learned about this subject. The interest in this topic will continue, keeping in line with the current trends in the industry. Of the various options outlined above, nutritional solutions are the most promising, cost-effective, and preferred. Despite the strong genetic component in GIT growth, enzyme secretion, and immunity, evidence clearly demonstrates that dietary factors can manipulate early development [43]. Feeding of the hatchling needs be designed for early growth and functionality of the GIT, promotion of early feed intake, stimulation of GIT health through development of commensal microbiome, and encouraging immune functions. The quote “well begun is half done (Aristotle, 384 BC–322 BC)” is appropriate here and any intervention strategy should begin immediately after hatch so that the broiler chick can cope with the enormous challenges in the rearing environment.

## Figures and Tables

**Table 1 animals-11-02795-t001:** Influence of broiler age on the total tract fat digestibility in three fat sources.

	Age (Days)	Fat Digestibility, %
Tallow	7	36.8
	14	65.3
	21	73.6
Soybean oil	7	59.1
	14	89.8
	21	96.5
Poultry fat	7	60.0
	14	84.5
	21	92.8

**Table 2 animals-11-02795-t002:** Total tract retention (% of intake) of minerals in broilers fed a maize–soybean meal-based diet during the first 14 days post-hatch.

	Day				
	3	5	7	9	14
Calcium	43	45	40	42	40
Phosphorus *	60	55	47	49	49
Potassium *	49	38	34	35	30
Sodium *	95	68	66	63	68
Magnesium *	39	29	26	27	23
Iron *	34	20	21	24	21
Manganese *	25	13	11	17	11
Zinc *	28	13	10	13	0
Copper *	23	12	8	9	4

* Significant age effects (*p* < 0.05).

**Table 3 animals-11-02795-t003:** Changes in the nitrogen-corrected apparent metabolizable energy (AME_n_; MJ/kg dry matter), nitrogen retention (% intake), and total tract digestibility (%) of starch and fat in maize–soybean meal-based diets for broiler chicks during the first 21 days posthatch.

	Day						
	2	4	6	8	10	14	21
Nitrogen-corrected AME	14.46 ^a^	12.91 ^b^	11.93 ^c^	12.09 ^c^	12.11 ^c^	13.22 ^b^	13.08 ^b^
Nitrogen retention	0.821 ^a^	0.717 ^b^	0.699 ^b^	0.635 ^c^	0.578 ^d^	0.638 ^c^	0.625 ^cd^
Starch digestibility	-	96.2 ^a^	-	93.6 ^b^	-	97.5 ^a^	-
Fat digestibility	-	68.7 ^b^	-	65.1 ^b^	-	77.5 ^a^	-

^a,b,c,d^ Different superscripts in a row are significantly different (*p* < 0.05).

**Table 4 animals-11-02795-t004:** Influence of age on the fat-free tibia ash content (% tibia) of broilers fed maize–soybean diets containing recommended concentrations of calcium and non-phytate phosphorus (Massey University, unpublished data).

Age (Days)	Fat-Free Tibia Ash. %
1	28.7 ± 3.90
7	42.3 ± 2.67
14	51.3 ± 2.69
21	50.0 ± 3.56
28	49.1 ± 2.94
35	51.5 ± 2.63
42	49.3 ± 2.75 ^1^

^1^ Mean ± standard deviation of 6 replicates.

**Table 5 animals-11-02795-t005:** Some aspects for consideration in the formulation and manufacturing of special pre-starter diets.

High-quality and highly digestible protein sources
Good-quality raw materials, with minimal deleterious factors
Higher than recommended amino acid densities
No saturated fats
Good fat quality
Fish oil to boost immunity
Optimum feed particle size to promote gizzard development High-quality mini-pellets or crumbles (low fine particles) to minimize selective feeding and promote feed intake Biotechnological processing (microbial fermentation and enzymatic pre-digestion) of feed ingredients to eliminate anti-nutritional factors [113]
Feed additives that promote commensal gut flora, e.g., probiotics
Feed additives that improve nutrient digestion or ameliorate the adverse effects of antinutrients, e.g., emulsifiers, exogenous enzymes
Increased sodium level to promote feed intake and nutrient absorption [114]
Minimal use of ingredients that cause inflammation
Higher levels of specific AA, e.g., glutamine, glycine + serine
Higher levels of specific trace minerals, e.g., zinc

**Table 6 animals-11-02795-t006:** List of feed additives useful in young chick nutrition.

Additive	Examples	Reasons for Use
Enzymes	Xylanases, β-glucanases, phytase, protease	To overcome the anti-nutritional effects of arabinoxylans (in wheat and triticale), β-glucans (in barley), or phytate (in all plant feedstuffs) and to improve the overall nutrient availability and feed value
Emulsifiers/biosurfactants	Lysophosphatidyl choline	Emulsification and improved lipid digestion
Antibiotic replacers ^1^		
i. Direct-fed microbials	Probiotics	Provision of beneficial bacterial species such as lactobacilli and streptococci
ii. Prebiotics	Fructo-oligosaccharides (FOS), mannan oligosaccharides (MOS)	Binding of harmful bacteria
iii. Organic acids	Propionic acid, diformate	Lowering of gut pH and prevention of the growth of harmful bacteria
iv. Botanicals	Herbs, spices, plant extracts, essential oils	Prevention of the growth of harmful bacteria
v. Antimicrobial proteins/peptides	Lysozyme, lactacin F, lactoferrin, α-lactalbumin	Prevention of the growth of harmful bacteria
Synthetic AA	DL-methionine, L-lysine, L-threonine	Diet formulation based on digestible AA and ideal protein concept

^1^ Due to the ban or restriction on the use of in-feed antibiotics, a multitude of compounds (individually and in combination) are being tested/used to improve the GIT health.

## Data Availability

Not applicable.

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
