# Peer review of "Nutrition and Digestive Physiology of the Broiler Chick: State of the Art and Outlook"

_animals, 2021, doi:10.3390/ani11102795_

Round 1
Reviewer 1 Report
Dear Authors,
The manuscript submitted for review is very extensive, contains a lot of valuable information from the literature review, and has been a valuable source of information in this important topic, which is: nutrition of the Newly Hatched Broiler Chick. The work is written in clear language that is understandable to the reader.
From the reviewer's duty, I would like to point out a few remarks:
- in general, the work is too extensive and some information is rather basic - textbook information, so I think that chapters 2 - 6 should be shortened by at least 30% - 40% and even combined into 1 or maximum 2, e.g. in chapter 4 the most important information is in the last 3 lines
- a possible solution is also to change the layout of the work and start the description with the current chapter 10 (without summary) and include the previous chapters (2-9) as sub-chapters but I leave the decision to the authors.
- table 4 contains unpublished data, or no comparable published data available? please explain
- why did the authors decide to resign for the description of the microbiome of the GIT? Please consider, if the paper will be shortened, than also could include some relevant information about the role of the microbiome,
Best regards
Author Response
The manuscript submitted for review is very extensive, contains a lot of valuable information from the literature review, and has been a valuable source of information in this important topic, which is: nutrition of the Newly Hatched Broiler Chick. The work is written in clear language that is understandable to the reader.
Response: Thanks for these comments
From the reviewer's duty, I would like to point out a few remarks:
- in general, the work is too extensive and some information is rather basic - textbook information, so I think that chapters 2 - 6 should be shortened by at least 30% - 40% and even combined into 1 or maximum 2, e.g. in chapter 4 the most important information is in the last 3 lines
Response: We appreciate the observations. Such a drastic shortening would have been possible in a research article by cutting down M&M, discussion etc. In a review, on the other hand, we believe that that all relevant background information must be provided for clear arguments, logical flow and coherence.
- a possible solution is also to change the layout of the work and start the description with the current chapter 10 (without summary) and include the previous chapters (2-9) as sub-chapters but I leave the decision to the authors.
Response: Responses as above. Moving Section 10 (summary) to the start and including the other current sections as sub-sections will be confusing and will not be the appropriate approach.
- table 4 contains unpublished data, or no comparable published data available? please explain
Response: To our knowledge, no comparable published data over the broiler growth cycle (hatch to 42 d) are available. Especially we wanted to emphasise the dramatic changes in tibia ash during the first 2 weeks.
- why did the authors decide to resign for the description of the microbiome of the GIT? Please consider, if the paper will be shortened, than also could include some relevant information about the role of the microbiome,
Response: This comment about ‘resign for the description..’ is not clear. The intention herein of mentioning gut microbiome only in relation to ‘early programming’ and ‘newly hatched chick’ and not to go into details, given the scant data during week 1.`

Reviewer 2 Report
This review article provides a review of the literature on digestive anatomy and physiology (gastrointestinal growth and development, maturation of the intestinal mucosa, secretion of bile and digestive enzymes, digestive passage rate and viscosity, digestion and nutrient utilization, development of skeletal system, nutrition of meat chicks (broilers) up to 14 days of age. The information presented is important for producers of broiler chickens, technology of production of feed mixes, scientific of veterinary, anatomy of physiology. Before publishing in Animals, the paper requires additions and corrections. The list of proposed changes is given below:
General comments:
Please prepare the article in accordance with the instructions for authors.
- Please provide the initials of the name and surname of each co-author of the article, the same as those given in the chapter "Author contributions"
- The email of each co-author and correspondent author must be this
- In the description of significance, please use lowercase p in italics, spaces before and after „<” . for example (p <05)
- The Author Conrtributions should contain the initials of the name and surname of the activities in accordance with the instructions for authors
- In the Reference section, abbreviated name journal must be revised and corrected, for example: Worlds Poult. Sci. J. instead of Wld's Pult. Sci. J. (item 9, 15, 20, 71, 93, 97, 102, 108, 130)
- In the reference chapter (item 3) 1991 instead of 1991a and others- without a liter, the publication number is enough to distinguish it from another source by this author in a given year
- In the References section, for a range of pages, use the long "-" from the insert function for all References items
Detailed comments
L2 I suggest: Nutrition and digestive system physiology of the broiler chick: State of the Art and Outlook
L42 digestive organs (add in parentheses)
L60-65 Replace with newer. The growth rate of broiler chickens in the last 3 decades has decreased. After hatching, the broiler chicks weigh approx. 42 g, while after 7 days of rearing - 175 g, increase in BW 19/day, over 300% for 7 days
L62% with data
L90 [14,15] with no spaces after 14
L105 + write something about the glycogen body in embryos
L99 Plesae, write about the differences in something in the microbiota of the upper and lower intestines
L132 [31-33] instead of the current form
L163 "relative” weights?
L200+ how about the length and diameter of small intestine segments? decisive and on the absorbent surface and nutrient absorption
L244 „researchers” instead of workers
L285 carboxypeptidase what about them?
L425, 466, 515 ”italics and small letters
L502 (Table 3) [35], Khalil et al. [88] instead of the current form
L523 with no spaces after day 7
L534 3-4% at what age?
L588 „injection” or „vaccines”?
L621-623 This is usually the case. However, hatching techniques with chick access to the feed in the hatcher have already been developed
L672 what toxins?
L674 what mycotoxins?
L714 What is the effect of AGP on Gut Health and percentage mortality?
L793 28, 35 and 42 days?
L975 no author - Tancharoenrat et al?
L1108 no page range
Author Response
this review article provides a review of the literature on digestive anatomy and physiology (gastrointestinal growth and development, maturation of the intestinal mucosa, secretion of bile and digestive enzymes, digestive passage rate and viscosity, digestion and nutrient utilization, development of skeletal system, nutrition of meat chicks (broilers) up to 14 days of age. The information presented is important for producers of broiler chickens, technology of production of feed mixes, scientific of veterinary, anatomy of physiology. Before publishing in Animals, the paper requires additions and corrections. The list of proposed changes is given below:
General comments:
Please prepare the article in accordance with the instructions for authors.
- Please provide the initials of the name and surname of each co-author of the article, the same as those given in the chapter "Author contributions"
Response: Initials provided in "Author contributions".
- The email of each co-author and correspondent author must be this
Response: Email address of both authors are added.
- In the description of significance, please use lowercase pin italics, spaces before and after „<” . for example (p <05)
Response: Corrected as suggested.
- The Author Contributions should contain the initials of the name and surname of the activities in accordance with the instructions for authors.
Response: Initials provided in "Author contributions".
- In the Reference section, abbreviated name journal must be revised and corrected, for example: Worlds Sci. J. instead of Wld's Poult. Sci. J. (item 9, 15, 20, 71, 93, 97, 102, 108, 130)
Response: Corrected as suggested.
- In the reference chapter (item 3) 1991 instead of 1991a and others- without a liter, the publication number is enough to distinguish it from another source by this author in a given year
Response: Corrected as suggested.
- In the References section, for a range of pages, use the long "-" from the insert function for all References items
Response: Corrected as suggested.
Detailed comments
L2 I suggest: Nutrition and digestive system physiology of the broiler chick: State of the Art and Outlook
Response: Revised as suggested
L42 digestive organs (add in parentheses)
Response: Revised as per suggestion
L60-65 Replace with newer. The growth rate of broiler chickens in the last 3 decades has decreased. After hatching, the broiler chicks weigh approx. 42 g, while after 7 days of rearing - 175 g, increase in BW 19/day, over 300% for 7 days
Response: Very good point. Revised and the suggested data are included.– See L 61-68.
L62 % with data
Response: Absolute weight data were not provided in these publications [3, 4], only the % gain changes
L90 [14,15] with no spaces after 14
Response: Corrected as suggested.
L105 write something about the glycogen body in embryos
Response: Revised. See L110-113
L99 Please, write about the differences in something in the microbiota of the upper and lower intestines
Response: Unfortunately, we are unable to add because do not see any data relating the chick during week 1
L132 [31-33] instead of the current form
Response: Corrected as suggested.
L163 "relative” weights?
Response: Corrected as suggested.
L200+ how about the length and diameter of small intestine segments? decisive and on the absorbent surface and nutrient absorption
Response: Good point. A statement is added. See L211-213
L244 „researchers” instead of workers
Response: Corrected as suggested.
L285 carboxypeptidase what about them?
Response: To our knowledge, no published data on this enzyme in young chicks is available
L425, 466, 515 ”italics and small letters
Response: Corrected as suggested.
L502 (Table 3) [35], Khalil et al. [88] instead of the current form
Response: Corrected as suggested.
L523 with no spaces after day 7
Response: Corrected as suggested.
L534 3-4% at what age?
Response: Good point. Information added. 3-4% during the first two weeks, then declining to ~2% at d35.
L588 „injection” or „vaccines”?
Response: Revised
L621-623 This is usually the case. However, hatching techniques with chick access to the feed in the hatcher have already been developed
Response: Agreed. Already indicated in this sub-section (Ref. 109)
L672 what toxins?
Response: Too many include in our manuscript – the relevant Ref#115 is cited - Revised
L674 what mycotoxins?
Response: Too many include in our manuscript – the relevant Ref#115 is cited -
L714 What is the effect of AGP on Gut Health and percentage mortality?
Response: Apologies – unable to provide any general figures as this is variable depending on existing conditions
L793 28, 35 and 42 days?
Response: Corrected as suggested.
L975 no author - Tancharoenrat et al?
Response: Yes - corrected as suggested.
L1108 no page range
Response: There is no page range in the journal.
